# Determination of Trace Lead and Cadmium in Decorative Material Using Disposable Screen-Printed Electrode Electrically Modified with Reduced Graphene Oxide/L-Cysteine/Bi-Film

**DOI:** 10.3390/s20051322

**Published:** 2020-02-28

**Authors:** Xiaopeng Hou, Benhai Xiong, Yue Wang, Li Wang, Hui Wang

**Affiliations:** 1Research Institute of Forestry New Technology and Research Institute of Wood Industry, Chinese Academy of Forestry, Beijing 100091, China; houxiaopeng_lunwen@163.com; 2State Key Laboratory of Animal Nutrition, Institute of Animal Science, Chinese Academy of Agricultural Sciences, Beijing 100193, Chinawangyue9313@163.com (Y.W.); 3Geographic Information Center of Yulin City, Shannxi 719000, China; rnjwl612@163.com

**Keywords:** sensor, heavy metals, bismuth film, DSPE, L-cysteine, decorative material, reduced graphene oxide

## Abstract

Cadmium (Cd) and lead (Pb) in decorative materials threaten human health. To determine the content of Cd(II) and Pb(II), a disposable screen-printed electrode (DSPE) electrically modified with reduced graphene oxide (rGO) and L-cysteine (LC) was fabricated, which was further electroplated with bismuth film (Bi/LC-rGO/DSPE) in situ. The electrochemical properties of this electrode were studied using cyclic voltammetry, electrochemical impedance spectroscopy, linear sweep voltammetry and differential pulse voltammetry. The results indicated that the Bi/LC-rGO/DSPE had excellent sensitivity, selectivity and stability with low cost and easy production. After optimizing the detection parameters, the linear range of the Bi/LC-rGO/DSPE was from 1.0 to 30.0 μg/L for Cd(II) and Pb(II), and the detection limits were 0.10 μg/L for Cd(II) and 0.08 μg/L for Pb(II). Finally, the Bi/LC-rGO/DSPE was applied to determine the concentrations of Cd(II) and Pb(II) in different decorative materials where the recoveries were in the range from 95.86% to 106.64%.

## 1. Introduction

Wood, as a renewable and recycled green material, is widely used in daily life. Due to its pore structure with hydrophilic groups [1], wood can absorb moisture, causing dry shrinkage, moisture expansion, warping, deformation, cracking or other defects [2]. To improve the practicability, surface decorative materials [3]—especially paint—are applied to coat the wood surface, which mainly prevents damage from moisture or scratches. However, some decorative materials contain soluble heavy metals [4]. These heavy metals can easily enter the human body through the skin or mouth [5] and threaten human health. In decorative materials, lead (Pb) and cadmium (Cd) are the two common pollutants, leading to some severe diseases [6,7], such as hypertension, cardiovascular disease, mental retardation and dementia.

To date, there is still no decorative material standard for heavy metals in China. Many developed countries, including the European Union, United States, Canada and Japan, have formulated relevant standards. According to the European Union Toy Safety Directive, the maximum contents of Pb(II) and Cd(II) are 90 mg/kg and 75 mg/kg, respectively, which is suitable to evaluate China’s decoration material industry. 

There are many methods to analyze the concentrations of Pb(II) and Cd(II). Traditional methods based on spectroscopic methodology, including atomic absorption spectrometry (AAS), ultraviolet–visible spectrophotometry (UV-Vis), atomic fluorescence spectrometry (AFS), inductively coupled plasma mass spectrometry (ICP-MS) and X-ray fluorescence spectrometry (XRF) have high sensitivity and selectivity [8]. However, these methods need expensive devices, a professional operator and complex sample pretreatment, so they can be applied in laboratories and professional testing institutions [9]. Therefore, it is highly urgent to develop a low-cost method to determine the concentrations of Pb(II) and Cd(II) that can conveniently analyze decorative materials in the home environment. Electrochemical stripping analysis (ESA) [10,11] is recognized as a powerful tool for the simultaneous determination of metal ions. In comparison to spectroscopic instruments, the relevant device is easy-to-use, portable, compact and inexpensive [12,13].

A disposable screen-printed electrode (DSPE) is a promising electrochemical electrode in view of its low cost, mass production and low background current. It can overcome some shortcomings of glassy carbon electrodes and carbon paste electrodes [14], including memory effects and tedious cleaning processes. In addition, the inherent disadvantages of DSPEs containing low sensitivity and reproducibility limit their practical application, but their surface is prone to modify different sensing materials to achieve excellent detection performance [15,16]. Therefore, DSPEs are suitable for Pb(II) and Cd(II) detection in the home environment. 

Graphene [17] is a two-dimensional carbon material with atoms arranged in a honeycomb lattice. Owing to its excellent physical, chemical, optical and electrical properties, it can be employed to decorate a DSPE’s surface to improve the sensitivity. However, graphene hardly dissolves in water to form a homogeneous solution, making it difficult to modify the DSPE’s surface directly. Graphene oxide (GO) is a vital derivative of graphene that can dissolve well in water because of the various oxygen-containing functional groups on the base and edge of graphene [18,19]. Since its discovery, graphene oxide has had unceasing applications in the field of sensors [20,21]. L-cysteine (LC) is a naturally occurring amino acid with good biocompatibility and water solubility. One LC molecule contains three functional groups (-SH, -NH_2_, -COOH) [22] that are easy to coordinate with the heavy metal ions. LC can be functionalized on the surface of electrochemical electrodes, which will increase the amount of Cd(II) and Pb(II) on the electrode surface [23,24,25,26]. Bismuth (Bi) is a low-toxicity element with low-temperature melting. It can form binary or multicomponent fusible alloys with numerous heavy metals, including lead, cadmium, thallium, antimony, indium or gallium, which can facilitate the nucleation process during the deposition of heavy metals [27]. 

Here, the DSPE was modified with reduced graphene oxide (rGO), LC and Bi film to improve the detection performance for Cd(II) and Pb(II). These three materials provided the DSPE with a large specific surface area, conductivity and high mechanical stability, which were investigated through different electrochemical methods. Finally, a sample pre-treatment method was established to extract the Cd(II) and Pb(II) from the decorative materials. 

## 2. Experimental

### 2.1. Reagents

GO was obtained from Nanjing Xianfeng Nanomaterial Technology Co., Ltd. Standard solutions of Bi(III), Cd(II) and Pb(II) (1000 mg/L) were provided by the National Standard Substances Center (China), which were diluted into the required concentrations. LC was offered by Shanghai Civil Chemical Technology Co., Ltd. A disposable screen-printed electrode was bought by Suzhou mayor’s triangle system, Interdisciplinary Research Institute Co., Ltd. Potassium ferricyanide and potassium ferrocyanide were offered by Beijing Chemical Industry Group Co., Ltd (Beijing, China). The supporting electrolyte obtained from Xiamen Haibiao Technology Co., Ltd. (Xiamen, China) was an acetate buffer solution (0.1 mol/L). Millipore-Q (18.2 MΩ cm) water was used for all experiments. The rest of the chemicals and reagents were of analytical grade without any further purification.

### 2.2. Apparatus

The electrochemical measurements were collected through a CHI 760 electrochemical workstation (CH Instruments, Austin, TX, USA). The conventional three-electrode system (a saturated Ag/AgCl reference electrode, a Pt wire auxiliary electrode and the Bi/LC-rGO/DSPE working electrode) was immersed into an electrochemical cell containing 10 mL of analyte liquor. During the deposition step, the analyte liquor was stirred using a magnetic stirrer (Shanghai Lichen Technology Co., Ltd, Shanghai, China). A PHS-3C digital pH meter was purchased from Shanghai Instrument Scientific Instrument Co., Ltd, Shanghai, China. All the experiments were performed at room temperature.

### 2.3. Electrode Preparation

The electrochemically reduced graphene oxide (rGO) and L-cysteine (LC) were modified on the DSPE’s surface in Figure 1 using the following protocol [9,23]. Firstly, the DSPE was rinsed with Millipore-Q water and dried in an N_2_ atmosphere. Then, 1 mg of GO was put into 5.0 mL of Millipore-Q water to form a homogeneous solution through ultrasonic treatment. The GO/DSPE was obtained by adding 3 μL of the dispersed solution on the DSPE’s surface directly. The GO/DSPE was heated at 50 ℃ for 60 min, and then immersed in a phosphate buffer solution (pH 7.0) to reduce the GO using cyclic voltammetry for 5 cycles with a scan rate of 20 mV/s. In addition, the rGO/DSPE was put into the 2.5 mol/L LC solution and scanned using cyclic voltammetry in the potential range of -0.4~+2.4 V with a scan rate of 40 mV/s. The modified electrodes (LC/DSPE and rGO/DSPE) were fabricated using parts of these steps. Bi can be corroded easily by oxygen in air or hydrolyzed in an alkaline and neutral environment, so bismuth film decorated on the surface of the LC-rGO/DSPE was usually carried out in an acidic environment. Thus, Bi film was deposited along with heavy metal detection.

### 2.4. Sample Extract Preparation 

The heavy metals were extracted from the decorative materials of the floor surface using the following protocol. Firstly, 1.0 g of decorative materials were scraped from the floor surface, and then ground to a powder. After that, the powder was added into 100 mL of a hydrochloric acid solution for 60 min. When the powder was deposited on the bottom of the bottle, the upper layer was transferred into another tube and filtered through a pore size of 0.22 μm. Prior to the measurement, the pH of the filter solution was adjusted to 4.5 using a 0.1 mol/L acetate buffer solution and 0.1 mol/L sodium hydroxide solution.

### 2.5. Measurement Procedure

The three electrodes were immersed in the acetic acid buffer or the sample extracts, and differential pulse voltammetry was applied to simultaneously measure Cd(II) and Pb(II). The LC-rGO/DSPE was enriched with the heavy metal ions at −1.2 V for 240 s by stirring and then equilibrated for 10 s. The parameters for the differential pulse voltammetry were as follows: amplitude, 0.004 V; amplitude, 0.05 V; pulse period, 0.5 s; sample width, 0.0157 s.

## 3. Results and Discussion

### 3.1. The Characteristics 

The electrochemical properties of the DSPE before and after it was decorated with reduced graphene oxide and L-cysteine were investigated by cyclic voltammetry (CV) and electrochemical impedance spectroscopy (EIS). The CV curves are shown in Figure 2 with a scan rate of 50 mV/s. The redox current of the bare DSPE was about 100 μA, and the potential between the oxidation peak and reduction peak reached 220 mV. On the LC/DSPE, the current redox response decreased to 72 μA with a peak-to-peak separation of 340 mV, ascribing that L-cysteine was a non-conductive material that prevented the transmission of electrons [28]. Moreover, the current redox responses of the rGO/DSPE were significantly higher than those of the DSPE and LC/DSPE. The reason was that the rGO provided a large specific surface area and electrical conductivity [29], which made the electrode surface have high electronic conductivity and accelerated the transfer rate of electroactive substances at the interface between the electrode and the solution. When L-cysteine was immobilized on the rGO/DSPE, the electrochemical performance of the LC-rGO/DSPE was decreased.

The equivalent circuit model of the LC-rGO/DSPE was employed to analyze the impedance spectroscopy in Figure 3. The EIS curve consists of two parts: a linear region at the lower frequencies and a semicircle region at higher frequencies, which represent the diffusion process and the electron transfer resistance (*R_ct_*), respectively. The *R_ct_* value of the commercial DSPE was about 1.25 kΩ. After L-cysteine was immobilized on the DSPE, the impedance between the LC/DSPE and electrolyte was increased, and the *R_ct_* value increased to 2.1 kΩ, which was ascribed to the poor chemical conductivity [30]. Due to the reduced graphene oxide, the *R_ct_* value of the rGO/DSPE was about 310 Ω. This small semicircle appeared on the rGO/DSPE, which indicates that the dynamic performance of the electronic transmission was poor. For the LC-rGO/DSPE, the *R_ct_* value increased to 700 Ω, which made the electroanalytical probe unable to reach the electrode surface to participate in the reaction. The results were in agreement with the conclusion obtained from the CV.

The hydrogen evolution has a profound effect on the electrochemical response. Hence, this property at the electrode surface was studied. As shown in Figure 4, the background current of the LC-rGO/DSPE had almost no change in the range from −1.0 V to −0.4 V and increases rapidly while the potential was lower than −1.0 V, indicating that hydrogen evolution occurred. It can be concluded that the electrochemical window of the LC-rGO/DSPE in the acetic acid buffer solution was about −1.0 V. After a bismuth film was deposited on the LC-rGO/DSPE surface, the hydrogen evolution potential of the Bi/LC-rGO/DSPE was shifted to the negative direction and reached about −1.3 V. This was mainly due to the formation of a bismuth film on the LC-rGO/DSPE, which expanded the electrochemical window [31]. Therefore, the Bi/LC-rGO/DSPE was much more suitable for the detection of Cd(II) and Pb(II) using the differential pulse anodic stripping voltammetric method.

Figure 5 shows the differential pulse voltammetry of 30.0 μg/L of Cd(II) and Pb(II) of the Bi/DSPE, Bi/LC/DSPE, Bi/rGO/DSPE and Bi/LC-rGO/DSPE. Two small peaks on the bare DSPE were located at −0.8 V and −0.6 V, which presented the stripping responses of Cd(II) and Pb(II), respectively. This can be attributed to the fact that bismuth can form an alloy with cadmium and lead that more readily reduces to Cd(II) and Pb(II) [32]. On the Bi/LC/DSPE, the stripping responses increased, which is mainly due to the -SH group that can bind strongly to heavy metal ions. The stripping responses of the Bi/rGO/DSPE were much higher than the Bi/DSPE owing to the high conductivity, large specific surface area and high adsorption capacity of the rGO. Furthermore, the stripping response of the Bi/LC-rGO/DSPE was better than that of the Bi/rGO/DSPE. This improved performance depended on the synergistic effects between the rGO and LC.

### 3.2. Analytical Performance

To investigate the analytical performance of the Bi/LC-rGO/DSPE, differential pulse voltammetry was used to determine different concentrations of Cd(II) and Pb(II) under the optimized parameters discussed in Appendix A (deposition time: 240 s; deposition potential: −1.2 V; supporting electrolyte: 0.1 mol/L pH 4.5 acetate buffer solution; Bi(III) concentration: 2 mg/L). Figure 6 shows the stripping voltammetric curves for a series of different concentrations of Cd(II) and Pb(II) under the optimized parameters. It was clear that the stripping responses of Cd(II) and Pb(II) had a positive correlation with the concentrations of Cd(II) and Pb(II), but the striping peaks were shifted to the lower voltage with increasing concentrations. For Cd(II) or Pb(II), the voltage of the initial stripping reaction was the same, and the reaction time changed with the concentration. The electrochemical reaction for the high concentration was longer than for the low concentration. With the scanning voltage of the differential pulse voltammetry (DPV), the stripping peaks shifted to the scanning direction of the DPV. Comparing by calculation, the stripping responses of Cd(II) and Pb(II) had a positive correlation with the concentrations of Cd(II) and Pb(II), which exhibited a linear relationship in the range from 1.0 to 30.0 μg/L. The linear regression equations and correlation coefficients were: $$\mathrm{Cd}\left(\mathrm{II}\right):\;y=0.6511\;x-0.615\;\left(R^{2}=9978\right)$$
$$\mathrm{Pb}\left(\mathrm{II}\right):\;y=0.5020\;x-0.3619\;\left(R^{2}=9951\right)$$

Based on the calculation of the triple signal-to-noise ratio (S/N = 3), the limits of detection of the Bi/LC-rGO/DSPE were 0.1 μg/L and 0.08 μg/L for Cd(II) and Pb(II), respectively. The proposed electrode can meet the requirements for Cd(II) and Pb(II) detection in the decorated materials.

Table 1 shows the properties of the Bi/LC-rGO/DSPE for Cd(II) and Pb(II) detection in comparison with the electrodes published in previous papers. The linear ranges and limit of detection are much lower than the modified glassy carbon electrodes but better than a majority of the modified screen-printed electrodes. The sensitivity is superior to most electrodes. Even though the Bi/LC-rGO/DSPE did not have excellent properties, it can meet the requirements for Cd(II) and Pb(II) detection in decorative material.

### 3.3. Interference Effects

The decorated materials might coexist with different metal ions (Na(I), K(I), Hg(II), Cu(II), Zn(II), Mg(II), Fe(III) and Al(III)), which were extracted at the sample pre-treatment stage. These metal ions probably interfered with the stripping responses of Cd(II) and Pb(II) on the Bi/LC-rGO/DSPE. The interference effects were studied through the Bi/LC-rGO/DSPE and explored against a standard solution containing 25 µg/L of Cd(II) and Pb(II) by adding 10-fold coexisting ions. In Figure 7, the highest relative currents were 2.07% and 3.61% for Cd(II) and Pb(II) when the Bi/LC-rGO/DSPE was determined as the standard solution adding 10-fold Hg(II). Therefore, different ions had no obvious interference, indicating the Bi/LC-rGO/DSPE has good selectivity. 

### 3.4. Application for the Sample Analysis

To evaluate the application, three samples of decorative materials were collected, which were pre-treated following the protocol in Section 2.4. The Cd(II) and Pb(II) concentrations in the extracted solutions were determined by the Bi/LC-rGO/DSPE and AAS. After each extracted solution was measured, the solution was added to 5.0 μg/L of Cd(II) and Pb(II), and then measured again. The results are shown in Table 2, indicating that the recoveries for Cd(II) and Pb(II) were in the range from 95.86% to 106.64%, and the relative standard deviations were less than 10%. This demonstrates that these two methods had no significant difference for Cd(II) and Pb(II) detection.

## 4. Conclusions

The DSPE was electrically modified with rGO, LC and Bi film for Cd(II) and Pb(II) detection with good stability. After the study, we found that the prepared electrode combined the advantages of the DSPE and the three materials, exhibiting excellent analytical performances, such as high sensitivity, stability (Appendix A) and anti-interference. The optimized parameters were pH 4.5, 2 mg/L of Bi(III) concentration, 240 s of deposition time and −1.2 V deposition potential. Under these conditions, the Bi/LC-rGO/DSPE had a wide linear response range from 1.0 to 30.0 μg/L for Cd(II) and Pb(II) with lower detection limits of 0.1 and 0.08 μg/L. The pre-treatment of the decorative sample was established, and the extraction was measured by the Bi/LC-rGO/DSPE. The recoveries for the Cd(II) and Pb(II) determination were in the range from 95.86% to 106.64%. Even though the established method possessed high precision and simple operation, the sample pre-treatment and detection times still need to be decreased.

## Figures and Tables

**Figure 1 sensors-20-01322-f001:**
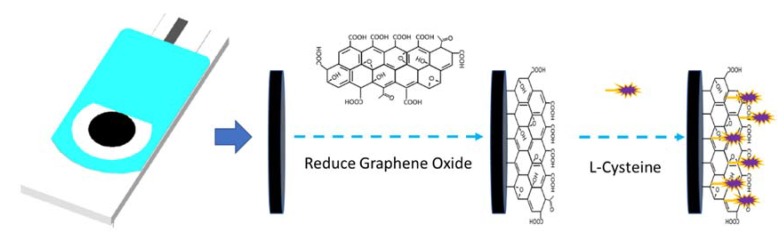
The process of a disposable screen-printed electrode (DSPE) modified to reduce graphene oxide and L-cysteine.

**Figure 2 sensors-20-01322-f002:**
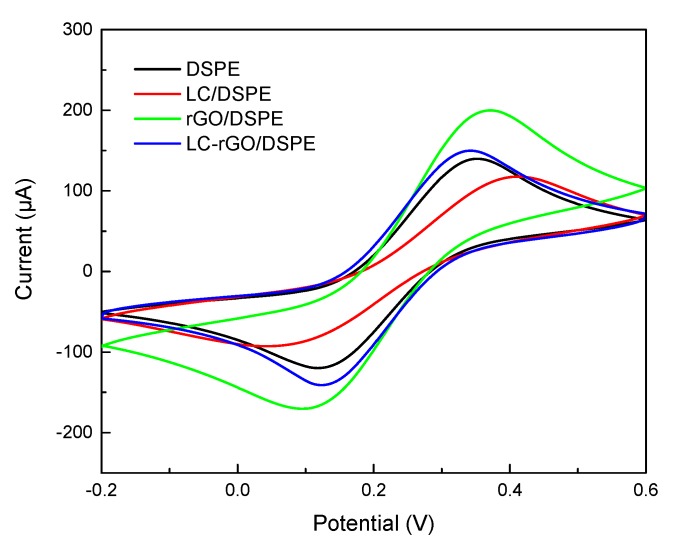
Cyclic voltammograms of the DSPE, L-cysteine (LC)/DSPE, reduced graphene oxide (rGO)/DSPE and LC-rGO/DSPE in 5 mmol/L [Fe(CN)_6_]^3−/4−^ containing 0.1 mol/L of KCl with a scan rate of 50 mV/s.

**Figure 3 sensors-20-01322-f003:**
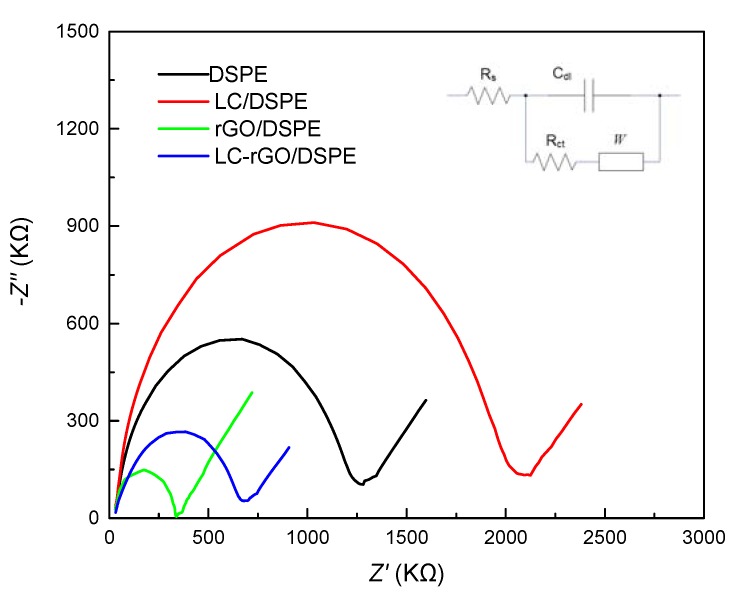
Electrochemical impedance spectroscopy of the DSPE, LC/DSPE, rGO/DSPE and LC-rGO/DSPE in 5 mmol/L [Fe(CN)_6_]^3-/4−^ and 0.1 mol/L KCl with frequencies from 1 to 10^5^ Hz.

**Figure 4 sensors-20-01322-f004:**
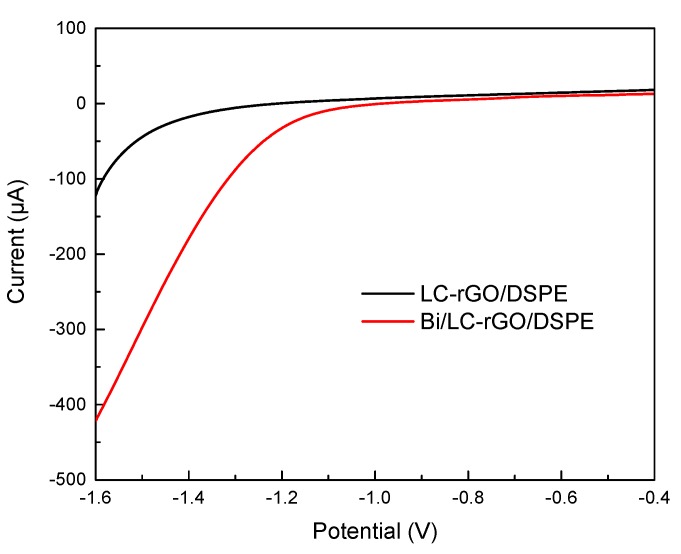
Linear sweep voltammetry of (**a**) the LC-rGO/DSPE and (**b**) Bi/LC-rGO/DSPE in 0.1mol/L acetic acid buffer solution with a scan rate of 50 mV/s (supporting electrolyte: 0.1 mol/L pH 4.5 acetate buffer solution; Bi(III) concentration: 2 mg/L).

**Figure 5 sensors-20-01322-f005:**
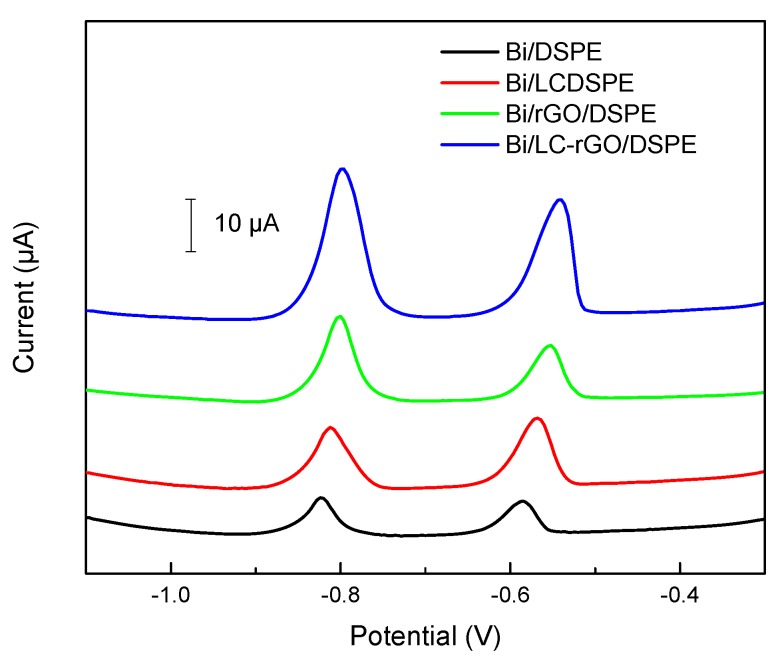
Differential pulse voltammetry of 30.0 μg/L of Cd(II) and Pb(II) by different electrodes: the Bi/DSPE, Bi/LC/DSPE, Bi/rGO/DSPE and Bi/LC-rGO/DSPE (deposition time: 240 s; deposition potential: −1.2 V; supporting electrolyte: 0.1 mol/L pH 4.5 acetate buffer solution; Bi(III) concentration: 2 mg/L).

**Figure 6 sensors-20-01322-f006:**
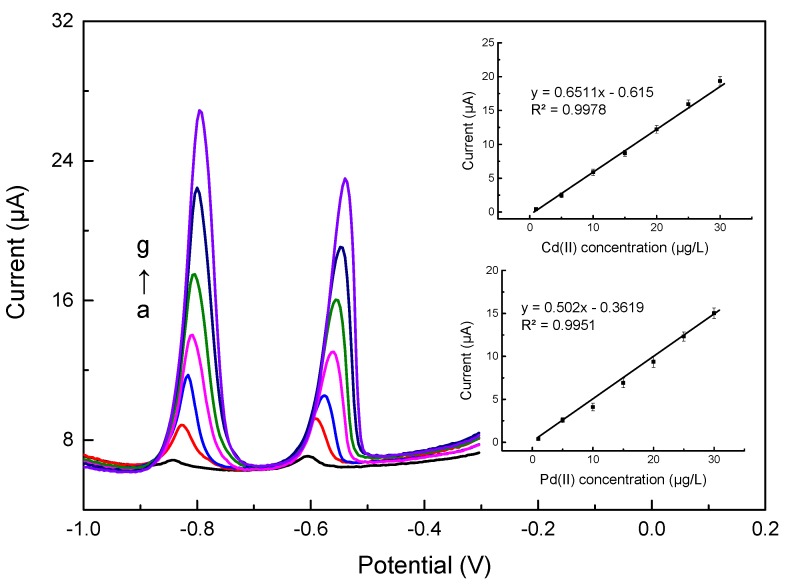
Differential pulse voltammetry of Cd(II) and Pb(II) on the Bi/LC-rGO/DSPE with various concentrations from a to g: 1, 5, 10, 15, 20, 25 and 30 μg/L. The inset shows the calibration curves for the determination of Cd(II) and Pb(II).

**Figure 7 sensors-20-01322-f007:**
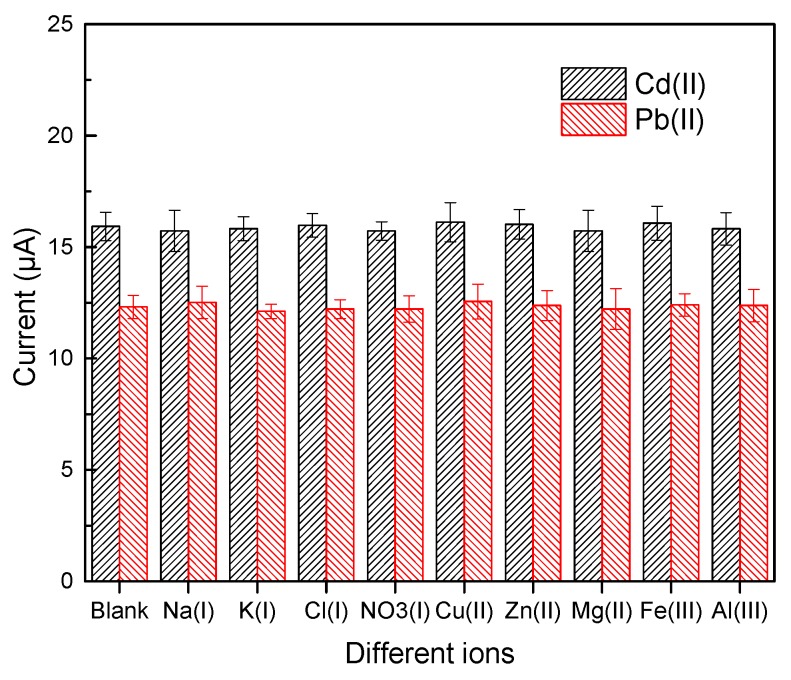
The selectivity of the Bi/LC-rGO/DSPE measured with 25 μg/L of Cd(II) and Pb(II) by adding 10-fold coexisting ions.

**Table 1 sensors-20-01322-t001:** The performances of different electrodes for Cd(II) and Pb(II) detection.

Electrode	Linear Range (μg/L)		Limit of Detection (μg/L)		Sensitivity (μA/(μg/L))		Reference
	Cd(II)	Pb(II)	Cd(II)	Pb(II)	Cd(II)	Pb(II)	
NFC/GCE	0.64–640	0.2–2000	0.32	0.11	0.64	0.13	[33]
Bi/Polyaniline/GCE	1.26–907	0.26–58.73	0.26	0.17	1.13	3.25	[34]
Bi/MWNTs/GO/GCE	-	5–40	0.31	0.21	-	0.06	[35]
Sb/SPCE	11.5–72.4	16.8–62.6	3.4	5.0	0.66	0.23	[36]
BiO/SPCE	0.5–12	0.5–12	0.2	0.2	0.83	0.67	[37]
G/PANI/SPCE	1–300	1–300	0.1	0.1	0.03	0.02	[38]
Bi/Au-GN-Cys/GCE	0.5–40	0.5–40	0.1	0.05	1.52	0.58	[39]
Bi/Nafion/PANI-MES/GCE	0.1–20	0.1–30	0.04	0.05	0.4	0.2	[40]
Bi/LC-rGO/DSPE	1–30	1–30	0.1	0.08	0.65	0.50	This work

NFC: Nanofibrillated Cellulose; Bi/MWNTs/GO: Bi film/Multiwalled carbon nanotubes/Graphene oxide; G/PANI: graphene–polyaniline nanocomposite; Sb: Antimony film; Au-GN-Cys: Gold nanoparticles/Graphene/L-Cysteine; Bi/Nafion PANI-MES: Bi film/Nafion/2-mercaptoethanesulfonate thiolated polyaniline.

**Table 2 sensors-20-01322-t002:** Results for the determination of Cd(II) and Pb(II) in the extracted solutions.

Sample	Add (μg/L)	Bi/LC-rGO/DSPE (μg/L) ^a^		AAS (μg/L) ^a^		Recovery (%)	
		Cd(II)	Pb(II)	Cd(II)	Pb(II)	Cd(II)	Pb(II)
1	-	1.57 ± 0.12 ^b^	3.24 ± 0.27 ^b^	1.63	3.38	96.32	95.86
	5	6.45 ± 0.52	8.54 ± 0.41	-	-	97.23	101.91
2	-	2.34 ± 0.47	4.67 ± 0.43	2.23	4.78	104.93	97.70
	5	7.46 ± 0.65	9.36 ± 0.69	-	-	103.18	95.71
3	-	5.65 ± 0.31	4.98 ± 0.31	5.59	4.67	101.07	106.64
	5	10.28 ± 0.53	9.81 ± 0.53	-	-	97.07	101.45

^a^ Atomic absorption spectrometry. ^b^ Value is the mean of three measurements ± standard deviation.

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
