# Peer review of "Determination of Trace Lead and Cadmium in Decorative Material Using Disposable Screen-Printed Electrode Electrically Modified with Reduced Graphene Oxide/L-Cysteine/Bi-Film"

_sensors, 2020, doi:10.3390/s20051322_

Round 1
Reviewer 1 Report
The manuscript describes the development of a combined lead and cadmium sensor using a screen-printed electrode modified with reduced graphene oxide/L-Cysteine and Bismuth. In general, the work is done properly and the authors have done a thorough characterization of the sensors. However, the English language requires major improvement, to the point that the reader doesnt even understand what the authors are trying to say.
Some specific points:
I suggest that the manuscript is proof read by a native speaker. There are a lot of grammatical and syntax errors in the text to the point that the reader doesnt even understand the content. The introduction section really needs improvements. The authors explain that cadmium and lead are added to PVC films and paints to prevent thermal degradation of PVC. PVC films are there to protect the wood from deteriorating due to moisture for example. At the same time, the authors claim that cadmium and lead from the PVC ends up in the human body by touching. The authors need to explain how can a sensor like this one be used to minimize the amount of cadmium and lead reaching the human body, if this is their application. If this is not their application, then they really need to provide the reason this sensor is really needed. I understand that this sensor is much lower cost and much easier to use that standard laboratory equipment. However, the authors need to explain that in the manuscript, not just state it. I believe that this issue arises from the fact that there is a lack of a good explanation of the application. The authors state that they are using a screen-printed electrode but they really need to provide the materials of at least the active layer of the electrode. Is it gold, platinum or graphene? Figure 3 shows the Nyquist plot of their EIS where the authors say that they used the equivalent circuit model. They really need to show that model as a circuit unless they used the classical Randles model where they need to mention it. Figure 10 shows the DPV results for both metals. It is interesting that the peak for each metal happens at a different voltage as the concentration increases. Do the authors have any explanation about this? The authors have done a good selectivity study but it would be very useful to show a percentage change on the current measured due to interference because although figure 11 is very good, it is difficult to see the difference when the absolute value is high. Also, in the text the authors say they used 30μg/L but in the caption they say 25.
Author Response
All the responses was in the attachemnt, please check it! The manuscript has been revised that marked different colors. If you want to know the details, you can check the revised manuscipt under the revision mode.

Reviewer 2 Report
The critical comments are as follows:
1. The abstract is descriptive and qualitative. Normally an abstract should state briefly the purpose of the study undertaken and meaningful conclusions based on the obtained results. Hence, this needs rewriting. I would expect brief, yet concise, the quantitative data description of the results in the abstract.
2. The level of English used is not up to the journal standard. Throughout the manuscript, the level of English used is not up to the standard of the journal. The sentences are long and badly worded with repetitive words. Please consider breaking longer sentences into smaller fragments for easy understanding. The authors are advised to seek help from a native English speaker. For example, the first four lines 15-18. Spilt into two sentences.
3. Introduction – unnecessary generality should be avoided for better readability and concise presentation. For example, Line 30 “Wood becomes the most attractive green material in recent years,” it is absolutely irrelevant to the theme presented in this study. Delete this sentence along with first two Ref. [1, 2].
4. Referencing is not right. Lines 67-74 lacks recent and relevant references. Add and discuss the following citation.
Hernandez-Vargas, G., Sosa-Hernández, J. E., Saldarriaga-Hernandez, S., Villalba-Rodríguez, A. M., Parra-Saldivar, R., & Iqbal, H. (2018). Electrochemical biosensors: A solution to pollution detection with reference to environmental contaminants. Biosensors, 8(2), 29.
Honeychurch, K. C., & Piano, M. (2018). Electrochemical (Bio) Sensors for Environmental and Food Analyses. Biosensors, 8(3).
Rasheed, T., Nabeel, F., Adeel, M., Bilal, M., & Iqbal, H. M. (2019). “Turn-on” fluorescent sensor-based probing of toxic Hg (II) and Cu (II) with potential intracellular monitoring. Biocatalysis and agricultural biotechnology, 17, 696-701.
Lazanas, A. C., Tsirka, K., Paipetis, A. S., & Prodromidis, M. I. (2020). 2D bismuthene/graphene modified electrodes for the ultra-sensitive stripping voltammetric determination of lead and cadmium. Electrochimica Acta, 135726.
5. Figures 6-9: reconstruct add error bars.
6. Table: What was the sample size? It needs to be clearly mentioned. Add a footnote explaining the coated values were taken from the duplicate/triplicate samples.
7. The conclusion section is barely a redundant expression of the abstract. Herein, I would like to see the major findings and how they are addressing the left behind research gaps and covering current challenges.
8. Referencing is not right and consistent. Many are missing vol. and page numbers. The authors should follow the journal guidelines for revised manuscript preparation. The reference list should be improved as there are many reports available from the year 2018-2020.
9. Editorial issues: The Latin names and Greek letters should be presented in italic in the whole manuscript, the unit presentation should be unified in the whole manuscript, abbreviations presentation should be unified.
Author Response

(The authors gave the same response as above.)

Reviewer 3 Report
This is an unremarkable piece of work, merely reporting the use of a carbon modified electrode for the electroanalysis of two trace metal ions. Screen printed electrodes, their modification and use, with Bi as a replacement for Hg, using stripping analysis have been known for decades. This contribution describes the application to the determination of Pb and Cd, the two metal ions which are the most simple and sensitive to be measured by DPP.
Author Response

(The authors gave the same response as above.)

Round 2
Reviewer 1 Report
The authors have done major modifications to the manuscript according to the reviewers' comments. I beileve that they have addressed almost all of the comments however, there are still some very minor changes needed:
1. My comment on figure 10 was not properly addressed since they authors did not fully understand my point. Of course, having peak voltages at -0.8 and -0.6V is because of the electrode potentials of different metals. My question is that for the same metal i.e. Cd(II) the peak is at approximately -0.8V. However, as the concentration decreaces, the peak voltage moves slightly to higher voltages. Do the authors have an explaination for that?
Author Response
Response: It will be easy to explain the reason from opposite direction. We can find that the initial voltage is almost at same voltage. When the concentration increase, the peak voltage moves slightly to lower voltage. The optimal stripping voltages for Cd(II) and Pb(II) were -0.8 V and -0.6 V, which also can strip in the small range of the optimal stripping voltages. The scanning velocity of DPV were constant. As for low concentration, the electrochemical reaction is completed in the initial time, so that the peak voltage occurs at the middle of this time. With the increase of concentration, the electrochemical reaction has to take a longer time, and the peak voltage occurs at the middle of this time. Therefore, the same initial reaction voltage and different reaction time can explain the peak voltages changing with the concentrations of Cd(II) and Pb(II).
Manuscript:
To investigate the analytical performance of Bi/LC-rGO/SPE, the differential pulse voltammetry was used to determine different concentrations of Cd(II) and Pb(II). Figure 10 shows the stripping voltammetric curves for a series of different concentrations of Cd(II) and Pb(II) under the optimized parameters. It was clear that the stripping responses of Cd(II) and Pb(II) were positive correlation with the concentrations of Cd(II) and Pb(II), but the striping peaks were shifted to the lower voltage with the increase of concentrations. For Cd(II) or Pb(II), the voltage of initial stripping reaction was the same, and the reaction time will be changing with the concentration. The electrochemical reaction for high concentration was longer than low concentration. With the scanning voltage of DPV, the stripping peaks shifted to the scanning direction of DPV. Comparing by calculation, the stripping responses of Cd(II) and Pb(II) were positive correlation with the concentrations of Cd(II) and Pb(II), which exhibited the linearity relationship in the range from 1.0 to 30.0 μg/L.
Reviewer 2 Report
The revised version reads well. Authors have addressed all the comments raised in the last review. This manuscript can now be accepted for publication.
Author Response
Thanks for your praise. We have further revised the article to highlight innovation.
Reviewer 3 Report
The authors have made some alterations and additions to the contribution but it remains low quality science. It would have been novel some decades ago but is now the routine reporting of the determination of two "easy" trace metal ions using standard equipment and technique, albeit with a minor electrode modification. For a paper of this length my opinion remains as "Reject". A much shorter contribution might be acceptable.
Author Response
Thanks for your advice. The English language and style has been revsed. In addition, the content was compressed, which has been moved to the supporting information. Please check the two files.